# Metallic Coatings through Additive Manufacturing: A Review

**DOI:** 10.3390/ma16062325

**Published:** 2023-03-14

**Authors:** Shalini Mohanty, Konda Gokuldoss Prashanth

**Affiliations:** 1Department of Mechanical and Industrial Engineering, Tallinn University of Technology, 12818 Tallinn, Estonia; 2CBCMT, School of Mechanical Engineering, Vellore Institute of Technology, Vellore 630014, Tamil Nadu, India

**Keywords:** additive manufacturing, coatings, metallic coating, powder bed fusion, direct energy deposition, spray coating

## Abstract

Metallic additive manufacturing is expeditiously gaining attention in advanced industries for manufacturing intricate structures for customized applications. However, the inadequate surface quality has inspired the inception of metallic coatings through additive manufacturing methods. This work presents a brief review of the different genres of metallic coatings adapted by industries through additive manufacturing technologies. The methodologies are classified according to the type of allied energies used in the process, such as direct energy deposition, binder jetting, powder bed fusion, hot spray coatings, sheet lamination, etc. Each method is described in detail and supported by relevant literature. The paper also includes the needs, applications, and challenges involved in each process.

## 1. Introduction

The advent of technology has led to an upsurge in the demand for more personalized products according to customer needs. In the age of the industrial revolution, there is a need for economically viable components without compromising the quality of several applications. The size and distribution of manufactured goods have challenged industries. Additive manufacturing (AM) is an emerging technology that provides flexibility in producing intricate parts at nominal costs, unlike conventional methods [1]. The technology has been filling the gap between conventional manufacturing methods and subtractive technologies for a decade. To explore the possible applications of AM processes, hybridization with conventional methods offers merged advantages. The higher material efficacy offered by AM technologies over the subtractive processes suppresses the expensive equipment costs [2]. Several attempts have been made by the research community to delve into different prospects of AM for metallic materials.

The idea of manufacturing parts through AM has been prevalent for a decade, and the applications include rapid prototyping, the generation of models for large-scale production, conducting different tests, and the validation of such models [3]. With the inception of the fourth industrial revolution, the direct manufacturing of products through AM emerged, including in the automotive, electronics, nuclear, aerospace, and bio-medical sectors. Unlike the available strategies such as servitization [4], presumption [5], and personalization [6], AM technology stands out as an effective manufacturing method among the leading industries. The AM process has gained popularity among researchers for the direct printing of parts, microstructure–property correlation, materials design, product design, and end utilization of the product [7,8,9]. Lately, AM has been used in conjunction with conventional manufacturing methods. Subtractive manufacturing accounts for controlled material removal from the substrate to obtain the final product, whereas in AM, layer-by-layer deposition takes place on the end surface [10]. As compared to traditional manufacturing methods, AM is intrinsically less harmful to the environment and leads to zero waste in terms of socio-economical value addition.

According to the American Society for Testing and Materials (ASTM International) standards, AM is classified based on the material used as feedstock, the state of fusion, the distribution of material, and the type of process [11]. As per ASTM standards, metal additive manufacturing (MAM) methods are broadly classified into direct energy deposition (DED)/powder-fed fusion (PFF) and powder bed fusion (PBF) processes. Apart from these, sheet lamination and binder jetting are also counted by ASTM as alternative MAM methods [12,13]. Some of the other potential methods include friction stir additive manufacturing [6,7], cold spraying [8,9], direct metal writing [10,11], and diode-based processes [14]. However, these techniques are still under consideration by ASTM to be included in the AM classification list. A generalized classification of the AM techniques is presented in Figure 1.

The metallic parts manufactured through AM processes often have irregular surface morphology when compared to those produced by conventional methods. The irregularities are the result of layer-by-layer deposition and fusion occurring on the material surfaces [15]. These drive the motivation to develop different alloy-rich layers or coatings on irregularly finished surfaces. The performance limitations of engineering materials for different applications have encouraged researchers to process them through AM coatings. In this regard, different AM methods are used in conjunction with the available conventional methods. Figure 2 indicates a broad outline of coatings on engineering materials through additive manufacturing technologies [16].

The factors that affect the irregularity of the surfaces include the staircase effect, the agglomeration of partially fused material, spattering, splashed particles (evaporation and balling effect), the instability of molten pool (wetting effect), etc. [17,18,19]. For instance, the addition of zinc in powder bed fusion processes enhances the wetting properties, whereas the uncontrollability in the movement of the molten pool at the boundaries is quite challenging [20,21]. Studies indicate the presence of irregularity in surface roughness on intricate geometries with inclined surfaces [22,23]. A change in inclination angle affects the surface roughness. The supporting structures when removed from the part geometry also alter the surface quality. The features formed on the surfaces become stress concentration sites for crack formation. In addition to this, the size of feedstock, deposition parameters, and surface morphology affect the surface morphology of the AM components [24,25,26]. For example, in the direct energy deposition (DED) process, the surface waviness is a result of the weld beads generated due to a large molten metal pool that is difficult to control [25]. Owing to the issues in the conventional AM processes, the interest among the research community in providing innovative solutions has accelerated. To minimize the manufacturing costs as well as to address the issues, AM methods were developed for coatings or surface modification. The urge to obtain diverse surface properties on a single component drove researchers to adapt additive manufacturing as a tool for coatings and surface modification. Different engineering materials are subjected to AM methods to modulate their surface morphologies and incorporate multiple properties in a single component. The present review article sheds light on the different AM methods adapted to carry out the deposition process. The advancements in AM coatings, technologies, challenges, and future opportunities are highlighted.

## 2. Need of Metallic Coatings

Thin film coatings usually possess a thickness of less than 0.1 μm, whereas thick film coatings have a thickness of more than 0.1 mm. As discussed earlier, coatings are made through AM to obtain enhanced properties on different components. Some of the significant advantages of coatings are as follows [27]:Ease of controlling the surface chemistry.Improving mechanical properties such as hardness, toughness, adhesion strength, etc.Inducing hydrophobicity or hydrophilicity to the surfaces.Enhancing anti-corrosive properties.Increasing bioactivities and improve biocompatibility.Improving tribological performance in terms of wear and friction.

## 3. Applications of Metallic Coatings

Some of the major functionalized applications of the AM metallic coatings are as follows:Aerospace, automotive, and missiles: parts to prevent loss in wear and corrosion.Automotive: brakes, bolted joints, etc.Electronics: fuel cells, sensors, MEMS/NEMS, field effect devices.Bio-medical: sterilization, cell adhesion, bio-implants such as pacemakers, and stents for dental application.Textile: self-cleaning fabrics, biofilms, anti-microbial surfaces, UV-protective materials (roofs, curtains, awnings, tents).Machine tools: cutting tools, electrodes, AFM tip, die, and molds.Power sector: turbine blades, heat exchangers, valves, and boiler parts.

## 4. Different Metallic Coatings through Additive Manufacturing

As discussed earlier, the classification of the metallic additive manufacturing methods is based on the type of allied energies used in the process. However, only a few of them are used for metallic coatings, some of which are described in the following sections.

### 4.1. Powder Bed Fusion

Powder bed fusion (PBF) includes the fusion of powders on the bed due to the introduction of a high amount of thermal energy. The high-dimensional stability obtained through this method whilst producing intricate and complex shapes makes it the most popular method in the metallic coating AM technologies. A wide range of printable powders (materials composition including Al-based [28,29,30,31,32,33,34,35,36,37,38,39,40,41], Fe-based [42,43,44,45,46,47,48,49,50,51], Cu-based [52,53,54,55,56,57], Ni-based [58,59,60,61,62,63,64,65,66,67,68,69,70], Ti-based [68,69,70,71,72,73,74], Mo-based [75,76,77,78], Co-based [79,80,81,82], Si-based [83], jewelry materials [84], etc.) can also be used in this method, which means components with multiple properties can be prepared. PBF is one of the popular methods commercially adapted from the group of AM technologies. Selective laser sintering (SLS) and selective laser melting (SLM) are the two most industrially acclaimed powder bed fusion processes that have been commercialized to date [12]. Some new technologies have also emerged, such as direct metal laser sintering (DMLS) [85], electron beam melting (EBM) [86], and laser curing [87].

The printing procedure for the PBF process is shown in Figure 3. The energy source (laser or electron beam) allows the fusion of powder particles after each layer feeding, thereby resulting in the formation of 3D structures (layer-by-layer deposition). However, prior to the printing process, preheating is required until the temperature is slightly lower than the glass transition temperature or the melting point of the powder. This cuts down the energy source power requirement during the printing process and expedites the fusion [12]. Moreover, preheating reduces the thermal gradient as well as the thermal distortion in the finished component [88,89,90]. Another important aspect of the process is that it needs an oxygen-free environment to undergo the fusion process. If there is a presence of oxygen in the chambers, the feedstock powders might oxidize before the actual start of the printing, which alters the final product’s surface properties. As an alternative to an oxygen-free environment, some inert gases such as nitrogen (for non-reactive powders), argon (for reactive powders), and vacuum (for electron beam) are used in the PBF process [91,92]. According to the type of energy source, PBF can be classified into laser-based PBF and electron-beam-based PBF.

#### 4.1.1. Laser-Based PBF

The laser PBF can be subdivided into selective laser sintering (SLS) and selective laser melting (SLM), the difference being the material preference and fusion mechanism of the powders [93]. On one hand, SLM melts the powder completely to form a homogeneous part, while SLS uses a point-based heating phenomenon and allows only molecular fusion at the surface. However, both processes can be used for coating purposes; SLM is preferred for the complete coating of surfaces whereas SLS is preferred for the localized deposition of engineering materials.

##### Selective Laser Sintering

Selective laser sintering (SLS) is a typical AM method wherein layer-by-layer deposition takes place by spreading the powders, followed by their selective sintering. Figure 4 is a typical schematic diagram of the SLS process constituting a powder layering setup, laser source, system interface, and other accessories (i.e., preheating unit and inert gas protection system). The types of lasers used in the SLS process include Nd:YAG [94], CO_2_ [95], fiber lasers [96], disc lasers [97], etc. The appropriate choice of lasers affects the strengthening of the powders for the following reasons:i.The laser absorptivity of the materials depends greatly on the laser wavelength.ii.The laser power energy determines the metallurgical changes occurring during powder densification.

The working procedure for the SLS process is as follows:The part to be fabricated is leveled and fixed on the platform bed.An inert-gas-filled atmosphere is created in the sealed building chamber to restrict the presence of oxygen during the process.Layering mechanism and laser beam scans enable the deposition of a thin layer of loose powder particles on the substrate material, allowing for selective molecular diffusion.The repetitive process of the above-mentioned steps helps in building the final part in a layer-by-layer fashion.

During SLS, the time of exposure of the laser beam depends on the scan speed and the beam size, which are usually 25 ms and 0.5 μm, respectively [98]. Owing to the need for a short thermal cycle, diffusion is the preferred means to combine the powders during the SLS process [99,100]. Partial melting of powders (or liquid-phase sintering) takes place in the process, and a semi-solid consistency of materials solidifies to be deposited on the substrate. SLS has recently demonstrated its efficiency in producing coated and alloyed components [101]. The metallurgical transformations taking place during the SLS process depend on the powder properties and the laser processing parameters. To obtain multiple attributes in a single component through SLS, the multicomponent powders used must possess a high melting point metallic component (acting as a structural unit), a low melting point metallic component (binder), and traces of additives, i.e., flux and deoxidizer [102,103]. The operative temperature for the SLS process is chosen between the two melting temperatures, and process parameters are fixed accordingly. The binder material completely melts into a liquid phase whilst the structural material retains its solid structural phase. The rearrangement of powder particles due to the capillary action of the wetting liquid results in the densification of the solid–liquid phase and the determination of the particle re-arrangement rate [104].

Zhu et al. [105] built multi-material components using pure Cu + SCuP (pre-alloyed) powders. SCuP powders act as binders owing to their lower melting point (i.e., 645 °C), whereas Cu takes the role of structural material due to its high melting temperature, 1083 °C, as shown in Figure 5a. In another study by Gu and Shen [106], a combination of higher-melting-point tungsten metal powders with copper metallic powders was fabricated through SLS. The densification behavior and the microstructural changes taking place during the SLS process were studied. Figure 5b shows the SEM images of the morphological changes occurring on the sintered samples at different line scan spacings. During melting and solidification, the pure metals with compatible melting points of pre-alloyed powders have a mushy (semi-solid) zone amidst the solidus and liquidus phases. With the optimization of the laser process parameters, it can be noted that the laser sintering temperature lies within the limits of the semi-solid zone, a process also known as supersolidus liquid-phase sintering (SLPS) [107]. As shown in Figure 6, the pre-alloyed power particles undergo extraneous melting and turn mushy as soon as an adequate amount of liquid is accumulated along the grain boundaries. The liquid wets the solid particles as well as the grain boundaries, thereby densifying the semi-solid system by the re-arrangement of the solid particles and precipitation of the solution. It is to be noted that the SLPS of the powders requires stern laser processing parameters to control the mushy zone. Localized and expeditious thermal cycles are generated during the laser sintering process. However, there are certain difficulties in controlling the laser sintering parameters, especially the temperature between the solidus and liquidus region, which seizes the SLPS mechanism. The problems encountered during the laser sintering operation, such as heterogeneity of microstructure, poor densification, an improper adaption of properties, etc., occur in the pre-alloyed powders. Thus, the post-processing of such parts manufactured through the laser sintering process needs careful attention. In this regard, several methods have been adapted to enhance the mechanical properties of the processed components. Some of the methods include hot iso-static pressing [108], furnace sintering [109], or secondary infiltration (in association with low-melting-point material) [110].

##### Selective Laser Melting (SLM)

The urge to obtain densified components with enhanced mechanical properties forced the research community to develop the laser melting process. The process stands out for its ability to produce parts with minimal post-processing cycles and material waste. SLM has the same procedure and apparatus as the SLS process. The only difference is that the complete melting and solidification of powders take place during the SLM process, whereas in the SLS process, just sintering takes place. The continuous improvement in the laser processing parameters, such as focused spot size, high laser power, smaller layer thickness, etc., has altered the metallurgical and mechanical properties [111]. As a result, SLM is best suited for producing parts with 99.99% relative density without any post-processing methods [112]. Li et al. [113] and Santos et al. [114] processed steel components using SLM and found that high scan speeds result in porosity in samples, along with a reduction in tensile strength. Figure 7 indicates the variation in porosity and microstructure in stainless steel samples prepared through SLM at different scanning speeds, wherein a significant difference in the morphology of the melt pool boundaries is visible.

Another advancement in the SLM process is its ability to process different categories of materials such as crystalline (high-entropy alloys), quasicrystalline, and amorphous systems [115,116,117,118,119,120,121,122,123,124,125,126,127], which is difficult through the partial melting SLS process. The earlier attempts to process the pure metals were unsuccessful through SLS, the reason being that the high viscosity of liquid material caused a balling effect that restricted the process [128,129]. On the contrary, the product manufactured through the SLM process is denser and can be controlled as desired [130,131]. However, SLM employs higher energy, which depends on laser power, the type of laser beam, exposure time, and layer thickness. Owing to such high energy input, instability in the molten pool may be observed, leading to a high degree of shrinking and internal stresses in the final component [24,132].

The residual stresses that arise during the SLM process due to rapid cooling also cause the distortion and/or delamination of the part. In a study conducted by Pogson et al. [133], it was affirmed that the incorporation of Cu into the tool steel imparts high energy input during the SLM process. This leads to the generation of austenite grain boundaries that might lead to cracking by hot tearing. The unstable melting may result in spheroidization of the melt pool, known as the balling effect, and can cause internal porosity in the samples. Some of the defects that arise during the SLM processes are shown in Figure 8. Therefore, suitable process parameters must be chosen to yield a moderate temperature, thereby avoiding the overheating of the SLM system [134].

#### 4.1.2. Electron-Beam-Based PBF

Electron beam powder bed fusion (E-PBF) uses a high-power electron beam to fuse metallic powder in a layer-by-layer manner into the final bulk component [93,139,140]. During the process, electrons emitted from the heated filament (W or LaB_6_) cathode are accelerated towards the substrate material at half the speed of light with 30–60 kV [141]. The process takes place under vacuum (He pressure < 1 Pa) to reduce the probable collision between the swift-moving electrons and the air molecules and to restrict the oxidation of metallic powders [142,143]. Electromagnetic lenses are used to focus the electron beam that can move at a deflection speed of about 10 km/s and a power of 5–10 mA, thereby allowing innovative heating and melting approaches. Once the electrons are bombarded to the powder bed, more of the kinetic energy is converted to heat energy, thus enabling local sintering of the powders. Figure 9 shows a schematic diagram of the E-PBF system. When compared to lasers, the electron beams penetrate significantly deeper (10^1^ to 10^2^) into the powder particles [144].

The powder bed is maintained at high temperatures, i.e., more than 870 K, and requires overnight cooling after job completion. E-PBF involves comparatively more processing parameters than LBPF technology. Some of the parameters include electron beam focus, power, scan speed, diameter, beam spacing, plate temperature, pre-heating temperature, contouring systems, and scan strategies [93]. The process parameter optimization is comparatively more difficult in the case of the E-PBF process than SLM; thus, only a limited number of materials can be processed through this method, i.e., CoCrMo [145], Ti grade 2 [146], Inconel-718 [147], Ti grade 5 [148], etc. The restrictions during the manufacture of intricate lattice structures (honeycomb) and low processing time make the process challenging. However, larger-sized products can be manufactured easily through this process irrespective of the substrate plate’s size. In a recent study, AlN coatings were produced over Ti6Al4V substrate through EB-PBF technology [149]. The authors achieved an adherent coating on the substrate without altering the core’s microstructure. The hybridization of EB-PBF with other methods such as chemical vapor deposition and atomic layer deposition allows for improved coating properties [150,151]. It is not advised to produce parts that constitute volatile components such as Mg, Zn, Bi, Pb, etc., through the E-PBF process. It is effective enough to process brittle materials, unlike SLM. The poor thermal expansion and contraction of intermetallic (brittle) materials tend to induce the formation of defects (solidification cracks) by restricting them to cool down at a slower rate. In this regard, SLM fails to slow down the cooling rate, thereby leading to crack propagation, while E-PBF allows the drop in cooling rates by increasing the temperature of the powder bed (~870 K) [152]. Thus, intermetallic materials such as TiAl and high-entropy alloys can be processed through the E-PBF process with careful consideration of temperature.

**Figure 9 materials-16-02325-f009:**
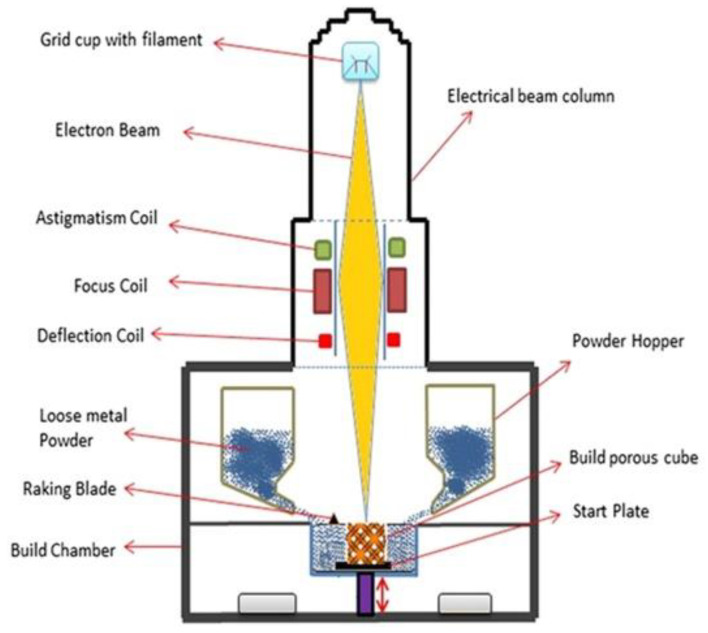
Schematic representation of the electron beam powder bed fusion technique [153].

### 4.2. Binder Jetting

Binder jetting is a type of AM technology that involves multiple steps for the fabrication of 3D parts. The process was introduced in the early 1990s by the Massachusetts Institute of Technology (MIT) and was commercialized in 2010 [154]. This technology can process almost all metals/alloys/ceramic materials (glass, graphite, sand, etc.) from powder form. The process requires a base material (metal/alloy/ceramic) of which the part is to be made, along with a binding agent (liquid phase) for gluing the material in layers. The printing technique is like any other AM printing process that takes place layer by layer. The powder (metallic/ceramic) material is spread over the bed by a roller according to a computer-aided design (CAD). The printing head allows the recurrent deposition of the binder adhesive as dictated by the CAD data over the powder bed [155,156]. As shown in Figure 10, the bed platform is adjusted or lowered based on the set layer thickness. As soon as the powder is bound to the binder adhesive, another layer of material is spread onto the previous layer, tending towards the final part. The loose powders that are unused or do not adhere to the layer surround the part until the final product is achieved.

Despite the simplicity of the binder jetting process, it involves several lengthy post-processing operations, such as sintering, de-powdering, curing, annealing, infiltration, and finishing [157,158]. One major advantage of this process is that there is no need for support structures while printing parts. The built parts remain on the powder bed without being bonded to each other. Thus, the entire volume of the built part can be stacked together, with many other parts to be printed with a small gap between them [159]. Because of the use of adhesives, this process is not recommended for structural applications (i.e., aerospace/automotive) since it might lead to porous parts. As compared to SLM/E-PBF, the binder jetting process is faster and can be further accelerated by implementing multiple printing heads/holes for deposition. It also allows multi-material deposition to obtain desired surface properties on a single component by changing the ratios of powder to binder. Coarser powders can also be used in this process, thereby cutting the manufacturing costs of finer powder particles. One more advantage of this process is the non-involvement of heat during the deposition process, thereby eliminating the formation of residual stresses in the final part [160]. Since the strengthening mechanism involved in the process is due to sintering, which may account for porosities, one may obtain varying shapes, volumes, and sizes of the pores in the final batch of products [161]. Furthermore, the final components are prone to having a coarse microstructure because of the post-processing operations. Thus, the parts produced through binder jetting lack suitable mechanical properties.

### 4.3. Direct Energy Deposition

#### 4.3.1. Laser-Based Material Deposition

Laser-based material deposition (LBMD) is a type of DED technique wherein the pre-spreading of powder on the bed is absent, and instead, coaxial feeding is executed. A schematic representation of the LBMD process is shown in Figure 11. In this method, the substrate is melted using the heat source (laser), thereby forming a melt pool that traps and melts the powder particles through a nozzle. The particles are driven away and mixed with a jet of gases (Ag or He). As the laser source is withdrawn from the bed, the molten material is solidified due to gradual heat dissipation. The nozzle head moves along the appropriate path (as dictated by the CAD model) and hence, the deposition takes place on the substrate [162]. After one layer of deposition, the nozzle head moves upward, and then again, another layer is formed. The previous layer acts as a substrate material for the next layer of deposition through the nozzle. The process of repetitive layer-by-layer deposition produces the final product [163]. The LBMD comprises DED and laser-engineered net shaping. Among all the laser-based AM technologies, LBMB has gained popularity owing to its ability to coat surfaces with ceramics or CMCs. The method is widely adopted for the surface modification of bulk materials to obtain enhanced properties, i.e., tribological, mechanical, chemical, or biological [164]. Due to the use of high-energy beam lasers, LBMD technology has the ability to process materials irrespective of their hardness at high melting temperatures [165]. The LBMD process thus has the ability to conduct selective surface modification, which can hardly be achieved through SLM and SLS techniques [166].

The repair and re-manufacturing of worn-out components are cost-effective when compared to buying new parts. LBMD has the potential to rebuild worn-out parts, which were previously considered un-repairable through conventional methods [167]. The process is most suitable for repairing turbine blades or vane tips with minimal distortion [12]. The closed-loop feedback and the vision system do not demand post-processing, thereby producing quality-based precision parts. The repair of driving shafts, bearings, couplers, and seals which are un-repairable by conventional welding methods can also be processed through the LBMD method [168]. The process facilitates metallurgical bonding between the deposition and the substrate, unlike other mechanical or chemical processes. In this regard, cladding and hard facing are also types of repair that build a protective or modified layer over the substrate through LBMD technology.

**Figure 11 materials-16-02325-f011:**
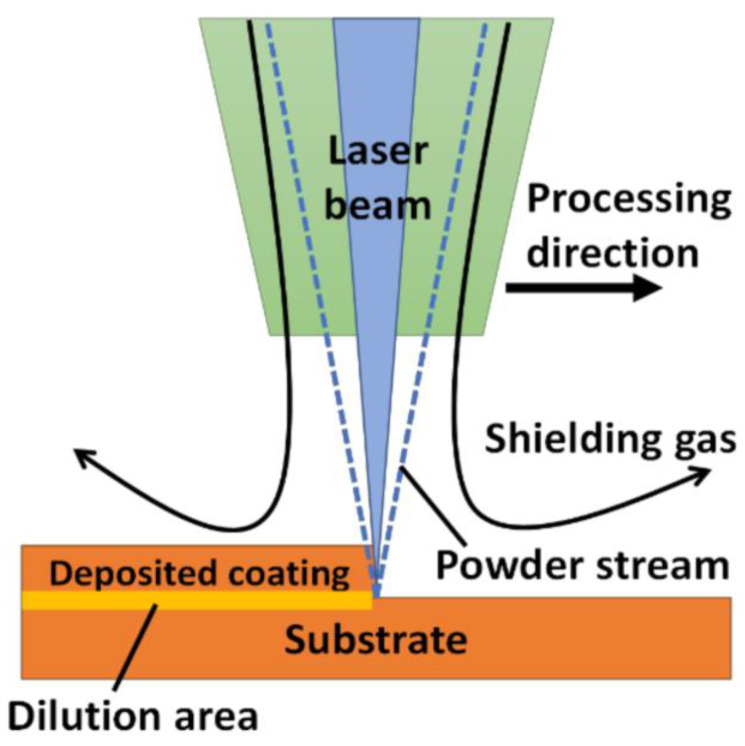
Schematic representation of the LBMD process [169].

Multi-layer coatings and surface modification can be carried out over complex geometries using the laser-based cladding technique [170,171]. In a study conducted by Kumar et al. [172], a hard coating of cBN was formed on a Ti6Al4V substrate using a laser cladding technique wherein hard phases of AlTi_3_N, TiN, and TiB_2_ were observed. From the microstructural analysis (Figure 12), they confirmed that fishbone (Figure 12a–c) and columnar (Figure 12d–f) structures and the presence of TiO_2_ nanoflakes reduce the crack susceptibility of the cladded area. Composite coating through laser cladding improves the surface properties in terms of tribological, mechanical, chemical, corrosive, and biological characteristics [173,174,175].

#### 4.3.2. Wire Arc Additive Manufacturing (WAAM)

Wire arc additive manufacturing (WAAM) is a wire-based AM technology that involves the direct deposition of weld beads in a layer-by-layer manner, thereby forming a metallic wall (minimum width: 1–2 mm). The wall formation is followed by building machining and then achieving a smooth surface [176]. The process looks like cladding carried out in subsequent melting and deposition of the wire feedstock over the substrate. This deposited part can be the final product or can be removed through conventional methods to obtain the final feature. The technologies that can be employed in WAAM are MIG [177], TIG [178], or plasma arc welding [179], as shown in Figure 13. One of the major advantages of the WAAM process is the low capital for initial investment since the machine assembling is sourced from welding industries [180]. Moreover, the processing characteristics of WAAM make it a preferable solution for other available fusion processes, as it does not use any vacuum environment, unlike the electron beam methods. Thus, the over-aging in precipitate-hardened materials can be avoided [181]. However, inert shielding is required in the case of WAAM to avoid contamination, whereas electron beam direct energy deposition does not require this [182,183,184]. The laser beam methods induce a high-power electrical arc as a source of fusion which is beneficial for reflective metal alloys (Mg, Cu, Al, etc.) [185]. The maximum layer height, roughness, and deposition rates that could be achieved through the WAAM technique are 1–2 mm, 500 μm, and 10 kg/h, respectively [186]. The advancement in WAAM technologies makes the processing of superalloys (Ti, Ni, Ta, etc.) easy.

The process is advantageous as it offers high deposition rates, the ability to manufacture intricate structures, and is adjustable with different torch movements and heat sources [188]. Nevertheless, the problems arising due to residual stress and distortion in WAAM, such as the welding or AM process, make the material processing challenging [186,189]. Ding et al. [189] contemplated surface finish as one of the concerns associated with the WAAM parts that lead to dimensional inaccuracy and premature part failures. Many attempts have been made to mitigate such issues but were limited whilst addressing the residual stresses. Pan et al. [190] discussed the mechanical properties of samples produced through WAAM along with the welding technology and process parameters used; however, they did not provide much detail on the mechanism. The work identifies an entire range of processing parameters, including heat treatment and inter-cooling procedures. Further research may explore the possibilities of using WAAM as a viable method for functional material grinding and generating parts with intricate designs.

### 4.4. Ultrasonic Additive Manufacturing

The ultrasonic AM (UAM) process is classified as sheet lamination AM by ASTM (ISM/ASTM52900-15, 2015). Unlike the solid-state joining (friction stir and ultrasonic metal welding) process, UAM does not deal with welding; rather, high-frequency plastic deformation amidst the metallic foils assists the joining phenomena. Since mechanical forces control the UAM process, intricate structures cannot be manufactured. In this regard, UAM can manufacture parts with solid structures that can counterbalance the applied forces [191]. The manufacturing of heat transfer devices and embedded electronics is possible using UAM when embedded with the CNC stage (Figure 14). The process allows the incorporation of multiple properties into a single material through cladding, transition joining, and surface modification. The low processing temperature used in UAM allows direct assimilation of heat-sensitive electronics with the metal structures, such as sensors for health-monitoring applications.

A schematic representation of the UAM joining process is presented in Figure 15. During the process, the sonotrode (tool/horn) generates micro-asperities in the form of surface roughness. Subsequently, the asperities collapse, shear, and material deposition takes place over the interfacial zone (<10 mm). Ref. [191] claims the roughness occurs due to shearing action which develops over the recrystallized zone. One of the advantages of the UAM process is the minimal heat generation at the deformation zone since localized deposition takes place in this process. The temperature is around 150 °C for Al- and Cu-based alloys, which could be used for fast-response thermocouples [192]. The deposition process can be carried out in an ambient atmosphere, and thus, solidification microstructures are absent.

### 4.5. Other Methods

#### 4.5.1. Cold Spraying

One of the most viable technologies used for addressing the failures in turbine and compressor blades is cold spray additive manufacturing (CSAM). The process makes sure the underlying crystal structures are not altered. It can coat new layers with wall thickness ≥ 1″, such as manufacturing gears, by controlling the motion of the spray nozzle and the external motor drive [194]. In the CSAM process, the substrate is impinged with a stream of metallic particles accelerated (velocity = 300–1200 m/s) by a highly compressed supersonic jet of gas at a temperature within the melting point of the powders. Effective bonding can be obtained by adjusting the process parameters and defining the critical velocity [195,196]. The interfacial bonding phenomena proposed by the research community are (i) surface adhesion because of interface-instability-induced physical anchoring effects, (ii) partial melting and fusion of metallic materials in the severely deformed region, and (iii) the defects caused by the oxidic layers formed over the substrate and the particles [197]. The other viewpoints regarding the metallurgical bond and the mechanical inter-locking explain the metallic bonding in CSAM [198]. A schematic representation of the process is shown in Figure 16.

The figure shows that the compaction, deformation, and plastic flow of the sprayed particles remove the oxidic layer from the surfaces, thereby exposing a larger portion of the area for metallic interaction. The low temperature involved in the process prevents the pernicious effects of the oxidic phase transformation, decomposition, grain growth, etc. [199,200]. These many advantages make CSAM a promising technology for surface modification and coatings of a wide range of materials, including metal matrix composites [201,202,203]. As discussed earlier, CSAM coatings do not possess thickness constraints, a reason why CSAM is the most popular solid-state process for coating new parts or repairing structures with ease [204].

**Figure 16 materials-16-02325-f016:**
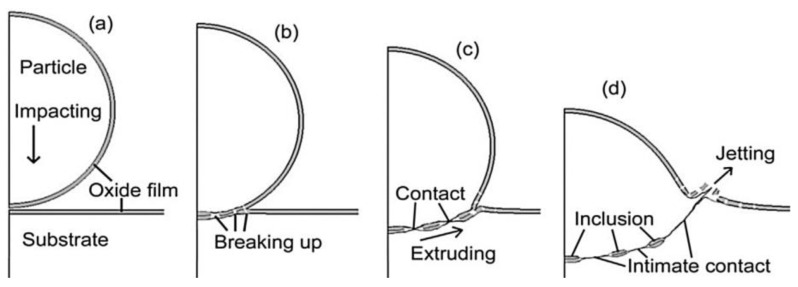
Schematic representation of the cold spray deposition process (**a**) impacting, (**b**) break up, (**c**) extruding, (**d**) inclusion [205].

#### 4.5.2. Magnetron Sputtering

The increasing demand to produce functionalized coated materials for diverse industrial applications has led to the emergence of magnetron sputtering. The process falls under the umbrella of physical vapor deposition (PVD). Deposition through sputtering is a complex method and at times, expensive [206]. However, the process allows controlling the composition during multilayer deposition and better flexibility in the type of material to be used [207]. Figure 17 shows the PVD reactors confined within a vacuum system with two electrodes passed through a high-voltage power source. A magnetron is placed in close proximity to the target while an ionic gas is flushed into the vacuum chamber, thereby bombarding the target with atomic-sized particles. Upon projection, these particles become stuck to the substrate and thus, deposition occurs.

The process allows the cleaning (cathodic cleaning) of previously contaminated surfaces and maintains potential differences between the target and the substrate [207]. Fewer stresses are developed on the substrate as the operating temperature for the deposition is within 50 °C [206,208]. Better film densification makes it a cleaner process [209]. However, the process yields a low deposition rate, film thickness, ionization efficacy in plasma, and issues with substrate pre-heating. These limitations led to the advancement of the process by introducing unbalanced magnetron sputtering, bipolar pulsed dual-magnetron sputtering, dual-anode sputtering, modulated pulsed power magnetron sputtering, etc.

#### 4.5.3. Electro-Spark Deposition

Electro-spark additive manufacturing (ESAM) or deposition is a type of low-energy (pulsed micro-arc) welding, better known for its minimal HAZ (heat-affected zones), low elemental diffusion, coatings with adequate metallurgical bonding, surface modification of conductive materials, and the ability to fuse dissimilar metallic materials [210]. Here, a high current and shorter pulse allow the melting of electrode material and droplet formation, and then droplets adhere to the substrate surfaces [211,212]. Proper optimization of process parameters (pulse frequency and movement of electrodes) generates multi-layer coatings of 5–10 μm for repairing purposes. With the inception of the initial deposited layer, a thin layer of mixed dilution is formed at the junction of the substrate and the coated region. The subsequently coated layers recede the effect of this mixed boundary layer, and thus, there is no contamination of the substrate with the electrode after the third layer of deposition [211]. A sharp thermal profile with narrow HAZ is generated at the interface when the molten droplet is transferred alongside the pulse discharge arc. Cathodic etching occurs on the base material, leading to metallurgical transformations, and the low mass transfer results in high cooling rates, minimal HAZ, and thermal stresses. Fine-grained microstructures are formed because of high cooling rates that curtail the elemental diffusion and formation of brittle inter-metallics [211]. The type of HAZ depends on the thermal characteristics of the substrate and the process parameters. ESAM imparts a low deposition rate because of the formation of discrete discharge that transfers a minute volume of molten material to the substrate surface. As the deposition thickness increases, the surface quality (roughness parameter) degrades, and the deposition rate decreases. Thus, to maintain the quality of the deposited layer, the surface roughness is maintained by grinding the samples using burr remover or sanding discs.

The process has been traditionally used for coatings and surface modification of metallic samples (tool steels, low- and medium-carbon steels, cast irons, die steels, cast steels, stainless steels, Al-alloys, Cu-alloys, and Ti-alloys) combined with ceramics. However, the coatings produced with high hardness and improved tribological properties need to be evolved. Irrespective of the discussed applications, ESAM is used for the post-processing operation of laser powder bed and fed processes. In a study by Enrique et al. [213], Inconel 625 samples prepared through binder jetting were subjected to AA4043 coating through the ESAM process using different parameters. Figure 18 shows the visuals of the coated surfaces produced through ESAM after the binder jetting of Inconel 625.

#### 4.5.4. Electrochemical Additive Manufacturing

Electrochemical additive manufacturing (ECAM) has gained popularity for objects having features of scale from mesoscale to nanoscale. Several applications are required to have features from the size of a few microns to 1 mm. Research [214,215] shows the relevance of these 3D-printed components for spatial applications. PCB manufacturing is one of the examples where there is feature size varying from the micron to mm range [216]. ECAM uses the principle of electrochemical cell reaction to print in metal at room temperature. An XYZ stage is required to carry out the localized 3D printing.
At anode (oxidation): M →M^n+^ +ne^−^ and Cu → Cu ^2+^ +2e^−^(1)
At cathode (reduction): Mn^+^+ne^−^ → M and Cu^2+^ +2e^−^ → Cu(2)

Here, the electrochemical cell reaction occurs in the presence of an electrolyte when the electrodes are supplied with a rated current. The tool head of the 3D printer holds the anode (positive electrodes) at which oxidation occurs to reduce the metallic electrode M to oxidize it into Mn^+^, as in the case of Cu, which is oxidized into Cu^2+^ with two electrons. Simultaneously, at the cathode (negative electrode), reduction takes place with the deposition of the metal ion Mn^+^, and acceptance of the negative electrode takes place, as in the case of copper Cu^2+^, which reduces to Cu by accepting 2e^−^, and the circuit is driven by the flow of electrons using an external power supply [215]. Figure 19 shows a schematic of an electrochemical 3D printer. Here, a syringe is employed to hold a certain concentration of electrolytes. The nozzle of a given diameter allows the flow of electrolytes and thus forms an electrolyte channel between the anode and cathode [215]. When supplied with DC electrical power supply, the redox reaction occurs at the junction, resulting in the deposition of the metal on the cathode. Using this method, localized deposition can be achieved, and thus, the process can be automated to print the features [217].

## 5. Materials Suitable for Metallic Coatings

The metallic coatings produced through additive manufacturing technologies are not limited to high-end alloys, i.e., Ti6Al4V [153,218], stainless steels [112,156], AlSi10Mg [219,220], Maraging steels [221,222], CoCrMo [145], and Ni-based alloys (IN718, IN625, etc.) [213,223]. New candidates are emerging to make the processes readily available to end users. In addition to the above-mentioned materials, some precious metals (gold, platinum, and silver) have been expanding into the AM market through SLM technology. Various factors control the use of limited engineering materials according to the extent of their weldability and castability to be used in different AM technologies [224].

## 6. Challenges in Coatings through Additive Manufacturing

The inception of different MAM technologies has benefited different industries, but there are several issues associated with these methods. The challenges can be categorized based on material and/or processing issues. An extensive discussion of the major issues in manufacturing metallic components is presented in this section.

### 6.1. Material Compatibility Issues

A wide spectrum of materials is utilized using MAM processes, including pure metals, alloys, ceramics, and composites, and further development of new materials is in progress [7,8]. The compatibility of the materials with a certain AM process is based on their mechanical and chemical properties along with their manufacturing ability. With dissimilar metal joining, mixing, or alloying, there is a chance of the formation of brittle phases, induced stresses, and interfacial defects [225,226,227]. When designing parts with multiple materials, the performance of the final components is of utmost interest. The process itself may have disadvantages such as dimensional accuracy, the need for post-processing operations, inability to address certain combinations of materials, difficulty in altering the working environment, and so on. The formation of an intermetallic bond layer while designing parts with different powder combinations involves complexity in the mixing and distribution of the materials [60,228,229]. Therefore, the selection of a compatible bond material is essential to fabricate parts with minimal or zero imperfections. In this regard, the study of the alloy phase diagrams is the baseline for comprehending the alloy compositions and material compatibility. A ternary phase diagram indicates the multiple equilibrium regions of different elemental compositions and eutectic temperatures [16,227]. These regions are quite difficult to assess in terms of determining relevant alloy compositions that will avoid the formation of brittle intermetallic phases. Therefore, sensible knowledge of materials science is desirable for envisioning ternary phase diagrams and equilibrium crystallization. Moreover, there is a need to expand the scope of a comprehensive database for specific materials, such as super alloys, that would be compatible with other available materials.

### 6.2. Defects, Flaws, and Dimensional Stability

Defect formation is one of the obvious issues in metal AM processes. Although the metal AM process is a mature technology owing to its ability to produce multi-material components with multiple inherent surface properties, the printed part is never free from interfacial defects. The commonly found defects in single-material-printed parts are micro-cracks, porosity, unmelted or solidified particles, etc. [16]. The issue with multiple-material printing is the compositional imbalances arising due to the metallurgical transformations occurring in the structures. A summary of the cause and type of defects formed in AM parts is presented in Table 1. Bonding defects occur due to insufficient energy input that might cause the generation of pores in the interface [162]. Cavities and micro-cracks are the results of shrinkage, while pores are a result of unusual gas filling in the molten material. In the case of deposition during cold spray AM technology, poor bonding between particles and inefficient plastic deformation may result in the generation of micro-pores and inter-particle boundaries at a lower impact velocity [3]. However, when the impact velocity is increased, the plastic deformation of powders is improved, thereby minimizing the inter-particle boundaries and the micro-pores. Another way to improve the plastic deformation is by heat treatment of the samples after the deposition process. Several heat treatment procedures have been adapted to introduce recrystallization along the inter-particle boundaries. Figure 20 shows the microstructural comparison between the Cu-deposited etched samples and heat-treated ones. The as-printed Cu deposition shows minimal pores and inter-particle grain boundaries. After the heat treatment (annealing), recrystallization occurs, wherein the particles assist in restoring the defects in grain boundaries (Figure 20b). Additionally, this repairing mechanism aids in improving the thermal and electrical conductivities of CSAM-deposited material [230].

### 6.3. Optimization of Process Parameters

The process parameters involved in the AM technology control the quality and performance of the produced component. Proper selection and optimization of the process parameters are necessary for addressing metallic materials, especially novel materials. In the case of the laser-based AM process, the user-defined parameters such as laser scan speed, feed, and power control the process [11]. Multi-objective strategies can be employed to fabricate parts with multiple powders. However, there exist certain challenges that control the selection of process parameters. The unavailability of proper standards is a major challenge in AM technologies. The processing parameters are either dependent on material, method, or machine [231,232,233]. For example, a laser-based process with the same metallic material and parameters will yield different results with varying machines. In the case of laser processing of Ti6Al4V through SLM, a higher scan speed and lower power are required as compared to the DED process. As per current industrial needs, SLM is the preferable AM technology over other available technologies; however, the processing time needs to be addressed [234]. Another challenge is the optimization of the process parameters, which requires a series of experiments. It is even more challenging to optimize the parameters for coatings through AM technologies. Owing to the dissimilarity in thermal properties between the substrate and coating, there is a chance that the interfacial region will generate flaws. To avoid these defects, proper optimization of the process parameters is necessary [177,235]. At present, optimizing the process parameters for coatings through AM technologies is a challenging task for the research community.

### 6.4. Environmental Hazard

The environmental impact is measured in terms of design for the environment, scoring systems, and lifecycle assessment. With the advancement in AM technologies, there is not much research evidence on the impact of environmental issues [225]. A joint effort of the designers, process control engineers, and environmental specialists is the need of the hour to assess environmental issues [236]. The environmental impact of laser-based AM technologies was discussed by Morrow et al. [237] through different case studies for quantitative analysis of energy and emissions. They concluded that laser AM technology has the potential to reduce the environmental impact alongside cost-cutting. The energy efficiencies of SLM and SLM processes were evaluated using cooperative efforts on process emissions in manufacturing [225]. The quantitative information on waste flows, emissions, and energy consumption is limited and needs in-depth exploration. Energy efficiency is often described as a ratio of output energy by the component to the total energy used up by the fabrication process [238]. The estimated energy efficiency of these processes accounts for 8.6% of the total energy consumed by the component itself. Although such information is valid and valuable, a holistic lifecycle assessment is often desirable.

The laser transfer energy efficiency ranges between 30% and 50% for fabricating parts of H13 tool steel and Cu powders. At optimal conditions, the deposition efficiency reaches 14% [239]. In one study [240], it was concluded that the SLS process is a sustainable AM system owing to its power consumption strategies, i.e., average power consumption = 19.6 kW, energy consumption by chamber heater = 36%, stepper motors = 26%, roller drivers = 16%, and laser source = 16%. The SLS process also has minimal waste and favorable energy indicators [241]. In addition to this, AM technology is a potential candidate for reducing carbon footprints through design optimization and waste management. As per the ATIKINS project, material and weight saving (by 100 kg) of almost 40% could be achieved for long-range aircraft [236]. There could be fuel savings of 2.5 MUSD and 1.3 Mt CO_2_ for an aircraft throughout its lifetime. Thus, the environmental aspect of AM technologies needs further exploration.

## 7. Conclusions and Outlook

Additive manufacturing is the most promising manufacturing method for processing materials with desired surface properties. This technology is moving towards value addition, sustainability, and cost optimization, thereby appealing to industries and the academic research community. To date, several AM technologies have been adapted for the manufacturing of metallic 3D complex features, but these are not free from limitations, including material compatibility, defects, dimensional instability, difficulty in incorporating desired properties, process parameter optimization, and environmental concerns. These flaws force the development of in situ monitoring processes and closed-loop process control to pre-qualify parts prior to post-processing operations and assembling. This aids in the further enhancement of AM technologies. The challenges of different AM processes for different applications have not been systematically detailed and require further research. Thus, there is a need for hybridization of the AM processes along with the implementation of new optimization methods, which might add value to industries and facilitate infiltration into the market. However, with the evolution of technology, challenges are evident. Therefore, the collective endeavor of scientists, researchers, engineers, and decision makers is necessary to iron out the technological, scientific, and economical issues associated with metallic additive manufacturing.

## Figures and Tables

**Figure 1 materials-16-02325-f001:**
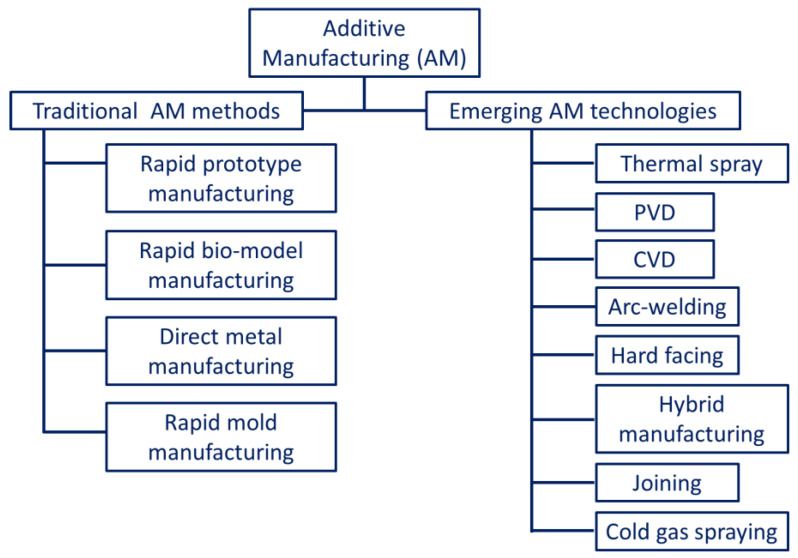
Generalized classification of the different additive manufacturing technologies.

**Figure 2 materials-16-02325-f002:**
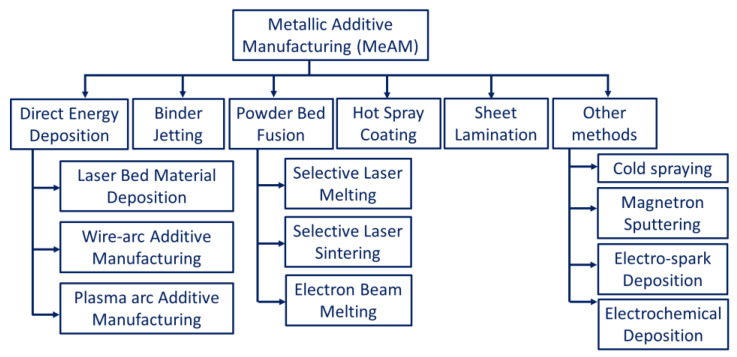
Broad classification of metallic additive manufacturing processes.

**Figure 3 materials-16-02325-f003:**
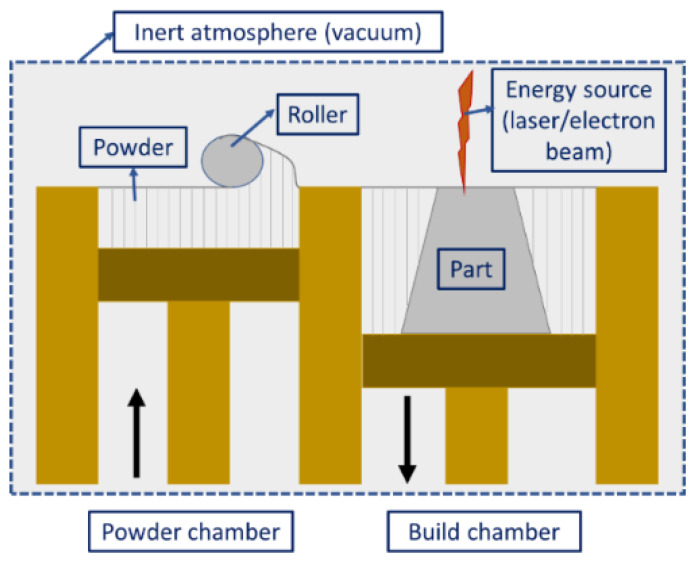
Schematic diagram illustrating the powder bed fusion process.

**Figure 4 materials-16-02325-f004:**
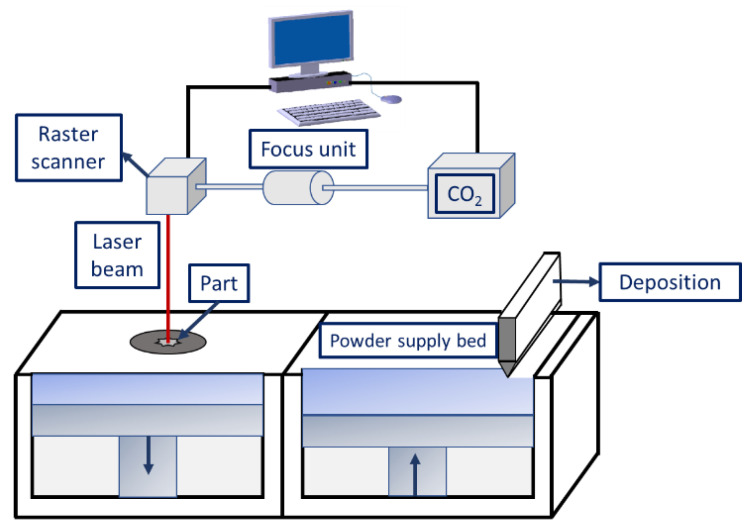
Schematic diagram illustrating the selective laser sintering process.

**Figure 5 materials-16-02325-f005:**
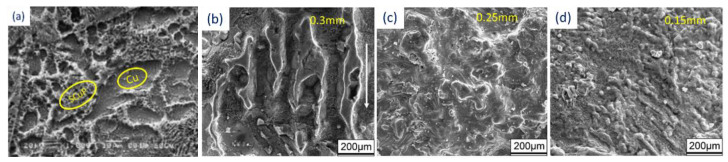
Scanning electron microscopy images of sintered samples prepared with (**a**) Cu-SCuP [105] and (**b**–**d**) W-Cu [106].

**Figure 6 materials-16-02325-f006:**
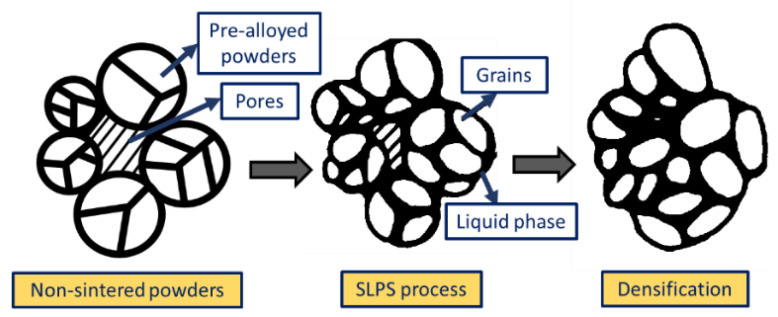
Schematic representation illustrating the densification process of the pre-alloyed powders during the SLPS process.

**Figure 7 materials-16-02325-f007:**
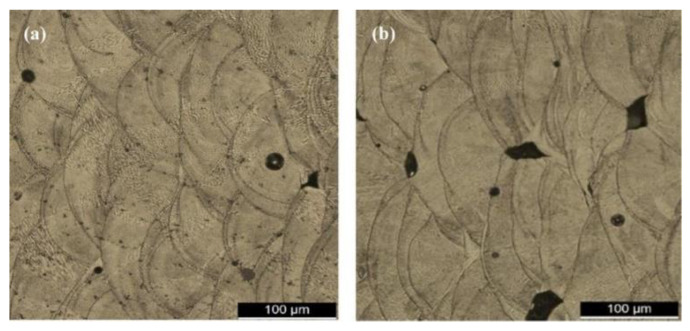
Microstructure showing the presence of porosity in stainless steel samples fabricated by the SLM process as a function of different laser scanning speeds: (**a**) 200 mm/s and (**b**) 400 mm/s [114].

**Figure 8 materials-16-02325-f008:**
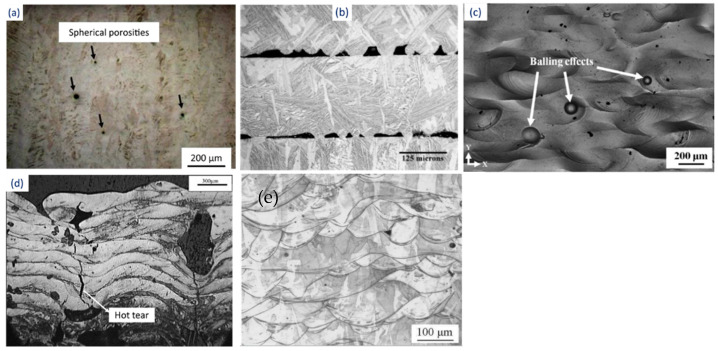
Defects generated during the SLM process: (**a**) porosity [135], (**b**) inadequate fusion between layers [136], (**c**) balling effect [137], (**d**) hot tear [133], and (**e**) fish scaling [138].

**Figure 10 materials-16-02325-f010:**
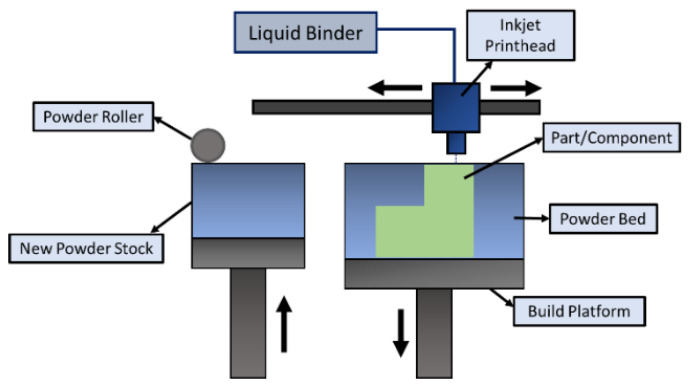
Schematic diagram illustrating the binder jetting process.

**Figure 12 materials-16-02325-f012:**
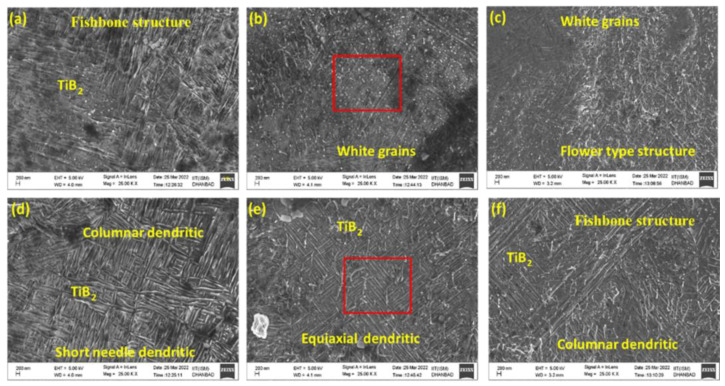
FESEM images indicating different microstructures on the prepared cladded samples showing (**a**–**c**) upper portion (**d**–**f**) middle portion [172].

**Figure 13 materials-16-02325-f013:**
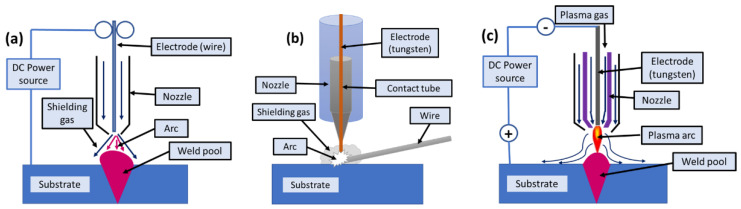
Wire arc additive manufacturing working principle using (**a**) MIG, (**b**) TIG, and (**c**) plasma arc welding [187].

**Figure 14 materials-16-02325-f014:**
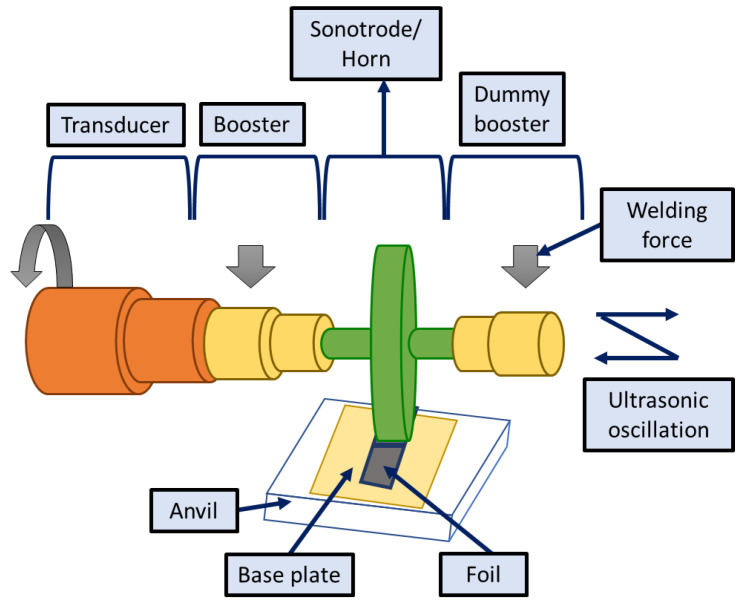
Ultrasonic additive stage illustrating the different components present.

**Figure 15 materials-16-02325-f015:**
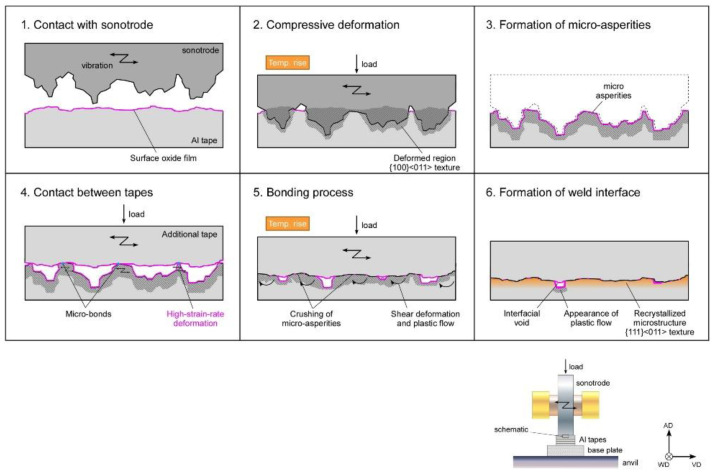
Schematic representation of UAM process for welding [193].

**Figure 17 materials-16-02325-f017:**
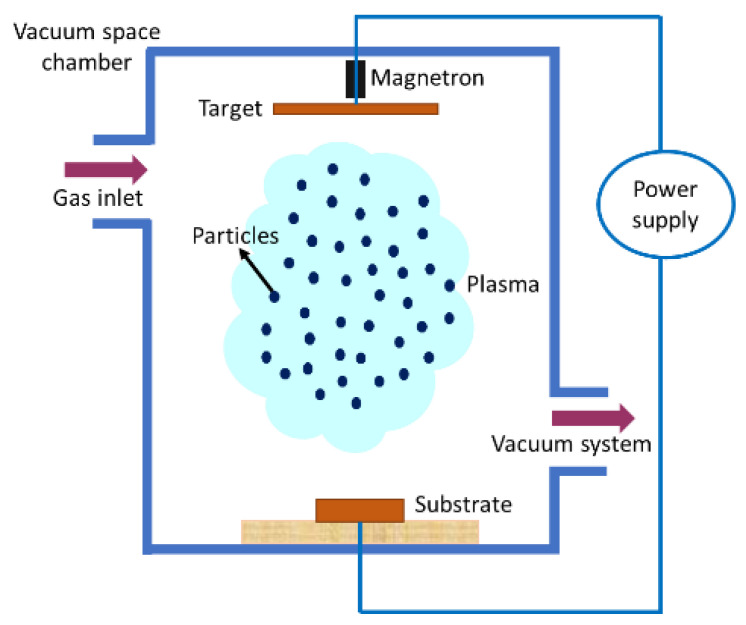
Schematic representation of magnetron sputtering process.

**Figure 18 materials-16-02325-f018:**
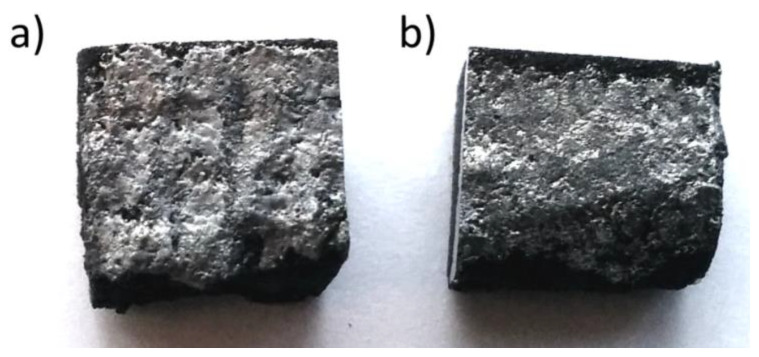
Post-processed Inconel 625-coated parts prepared though the ESAM process using the following parameters: (**a**) 600 mJ, 120 μF, and (**b**) 400 mJ, 80 μF after binder jetting [213].

**Figure 19 materials-16-02325-f019:**
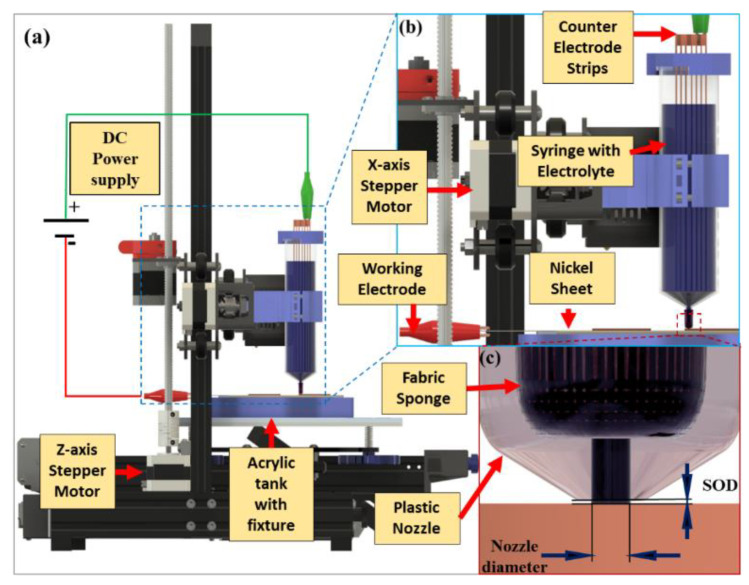
Development of lab-scale ECAM (**a**) setup, (**b**) syringe arrangement, and (**c**) nozzle tip [215].

**Figure 20 materials-16-02325-f020:**
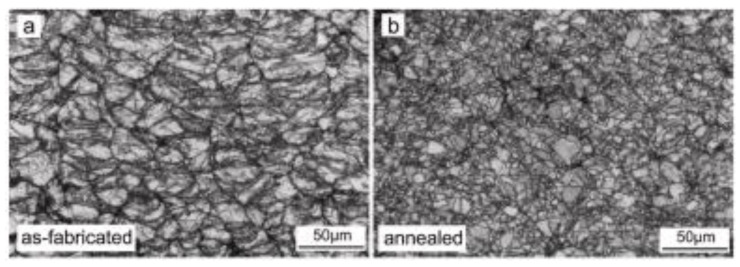
Comparison of microstructure of CSAM part (**a**) without and (**b**) with heat treatment (annealing) [3].

**Table 1 materials-16-02325-t001:** Some defects associated with AM parts.

Type of Defect	Cause of Defect
Micro-cracks or distortion	Differences in thermal gradient result in residual stresses, thermal expansion, and shrinkage during quenching cycles.
Gas porosities	Entrapment of gaseous molecules during powder atomization; shielding gas entrapment in the molten pool at a high powder flow rate; or moisture in the powders.
Roughened surfaces	Molten balls formed due to thermal gradients, unstable capillarity of the molten pool, hydrodynamic instability, spattering and denudation, splashing of molten material due to ejection, and change in the melt flow direction.
Lack of fusion and incomplete material melting	Insufficient energy passing through the powders and surfaces.
Keyholes	In fusion-based MAM processes, vaporization of constituent materials at high energy density.

## Data Availability

Not applicable.

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
