# Peer review of "Metallic Coatings through Additive Manufacturing: A Review"

_materials, 2023, doi:10.3390/ma16062325_

Round 1
Reviewer 1 Report
The author's manuscript needs to be reworked

Author Response
REVIEWER 1
Comment 1: The authors should make clear distinction between processes of thin and thick coating, cladding etc.
Thin film coating usually possesses a thickness not more than 0.1μm, whereas thick film coating is more than 0.1mm.
Comment 2: The authors seems to have equated AM processes and coating in most part of the manuscript. Can all the metal AM processes be used for coating/cladding? The authors should clarify with justification.
The authors agree with the reviewer that not all metal AM process can be used for coating/cladding. Thus, in the present review article, only limited metal additive manufacturing processes have been discussed.
Comment 3: Sections 2 and 3 needs more clarification, as all the points mentioned there can not be achieved using the AM-based coating processes. So, the authors should elaborate those aspects.
The authors have modified section 2 and 3 as per the suggestion and added more references to the bullets to support the statements.
Comment 4: The authors did not take the materials science aspect into consideration, and the types of materials which are suitable for AM- based coating must be discussed.
The authors are thankful to the reviewer for his/her suggestion. The material science aspect is not considered in the work as the authors wanted to present an overview of different AM technologies used for coatings, focussing more on the energy aspect. However, the authors would work on another article focussing on the material science aspect. Nevertheless, authors have included another section (section 5) showcasing the different type of materials that could be used for the metal additive manufacturing processes.
Reviewer 2 Report
The manuscript presents a state-of-the art review of the additive manufacturing processes for metallic coating. Overall, the manuscript in its current form appears more like a generalized review of different metal AM processes instead of a review of AM-based coating processes. The authors should address the following issues to improve the manuscript-
1. The authors should make clear distinction between processes of thin and thick coating, cladding etc.
2. The authors seems to have have equated AM processes and coating in most part of the manuscript. Can all the metal AM processes be used for coating/cladding? The authors should clarify with justification.
3. Sections 2 and 3 needs more clarification, as all the points mentioned there can not be achieved using the AM-based coating processes. So, the authors should elaborate those aspects.
4. The authors did not take the materials science aspect into consideration, and the types of materials which are suitable for AM- based coating must be discussed.
Author Response
REVIEWER 2
Comment 1: Lines 46 through 48" Subtractive manufacturing accounts for material removal to obtain the final product, whereas AM imitates the biological coating method wherein layer-by layer deposition takes place on the end surface." The description of the subtractive manufacturing was inappropriate.
The authors have revised the statement as per the suggestion.
Comment 2: In Figure 2 Broad classification of metallic additive manufacturing processes, duplicate classifications occur.
The authors are sorry for the error, the figure indicating the classification of metallic additive manufacturing processes has been replaced with a new one.
Comment 3: In " 2. Need of metallic coatings and 3. Applications of metallic coatings", more references may be needed.
The authors have added new references to section 2 and section 3 for each bullet, as per the suggestion of the reviewer.
Comment 4: In " 4.1.2. Electron-beam based PBF," These sections may need to be amended; add more research about metallic coatings through additive manufacturing.
The authors have added more reference as per the reviewer’s suggestion.
Comment 5: The sharpness of Figure 9, Figure 13, Figure 21
The authors have improved the images as per the suggestion.
Round 2
Reviewer 2 Report
It seems the authors have addressed the issues adequately, and so the manuscript can be accepted in its present form.